# Transcriptome Signatures of Atlantic Salmon—Resistant Phenotypes against Sea Lice Infestation Are Associated with Tissue Repair

**DOI:** 10.3390/genes14050986

**Published:** 2023-04-27

**Authors:** Valentina Valenzuela-Muñoz, Cristian Gallardo-Escárate, Diego Valenzuela-Miranda, Gustavo Nuñez-Acuña, Bárbara P. Benavente, Alejandro Alert, Marta Arevalo

**Affiliations:** 1Interdisciplinary Center for Aquaculture Research (INCAR), University of Concepción, Concepción P.O. Box 160-C, Chile; crisgallardo@udec.cl (C.G.-E.); divalenzuela@udec.cl (D.V.-M.); gustavonunez@udec.cl (G.N.-A.); bbenavente@udec.cl (B.P.B.); 2Laboratory of Biotechnology and Aquatic Genomics, Department of Oceanography, University of Concepción, Concepción P.O. Box 160-C, Chile; 3Centro de Biotecnología, Universidad de Concepción, Concepción P.O. Box 160-C, Chile; 4Hendrix Genetics Aquaculture, Villarica P.O. Box 4930000, Chile; alejandro.alert@hendrix-genetics.com (A.A.); marta.arevalo@hendrix-genetics.com (M.A.)

**Keywords:** *Salmo salar*, sea lice, *Caligus rogercresseyi*, skin, early infestation

## Abstract

Salmon aquaculture is constantly threatened by pathogens that impact fish health, welfare, and productivity, including the sea louse *Caligus rogercresseyi*. This marine ectoparasite is mainly controlled through delousing drug treatments that have lost efficacy. Therein, strategies such as salmon breeding selection represent a sustainable alternative to produce fish with resistance to sea lice. This study explored the whole-transcriptome changes in Atlantic salmon families with contrasting resistance phenotypes against lice infestation. In total, 121 Atlantic salmon families were challenged with 35 copepodites per fish and ranked after 14 infestation days. Skin and head kidney tissue from the top two lowest (R) and highest (S) infested families were sequenced by the Illumina platform. Genome-scale transcriptome analysis showed different expression profiles between the phenotypes. Significant differences in chromosome modulation between the R and S families were observed in skin tissue. Notably, the upregulation of genes associated with tissue repairs, such as collagen and myosin, was found in R families. Furthermore, skin tissue of resistant families showed the highest number of genes associated with molecular functions such as ion binding, transferase, and cytokine activity, compared with the susceptible. Interestingly, lncRNAs differentially modulated in the R/S families are located near genes associated with immune response, which are upregulated in the R family. Finally, SNPs variations were identified in both salmon families, where the resistant ones showed the highest number of SNPs variations. Remarkably, among the genes with SPNs, genes associated with the tissue repair process were identified. This study reported Atlantic salmon chromosome regions exclusively expressed in R or S Atlantic salmon families’ phenotypes. Furthermore, due to the presence of SNPs and high expression of tissue repair genes in the resistant families, it is possible to suggest mucosal immune activation associated with the Atlantic salmon resistance to sea louse infestation.

## 1. Introduction

Salmon aquaculture is constantly threatened by pathogens that impact fish health, welfare, and productivity. The most prevalent parasite disease that affects Chilean salmon aquaculture is Caligidosis, caused by the marine ectoparasite *C. rogercresseyi* [1]. The economic losses from this disease were estimated to be around USD 463 MM in Chilean salmon farms [2]. For sea lice control, the most used method is pesticides or delousing drug treatments. However, due to their intensive use, lice have lost sensitivity to pesticides [3]. Therein, strategies based on salmon breeding selection represent a sustainable alternative to produce fish resistant to sea lice infestation.

Genomic Selection (GS) tools have been widely used in animal selection programs [4]. These studies rely on identifying genetic variations among individuals statistically associated with a specific trait of interest. Here, single nucleotide polymorphisms (SNPs) have mainly been used as genetic markers in GS studies to calculate genomic breeding values without prior knowledge of the underlying [4,5]. The GS markers are estimated in “training” populations, which have been measured for contrasting phenotypes, for instance, susceptible and resistant individuals to a disease [5]. The advantage of this method is the high prediction occurrence. For instance, GS tools have shown a genetic component in fish resistant to the salmon louse *Lepeophtheirus salmonis*, with heritability values of 0.2–0.3 in Norway and Canada [6,7].

On the other hand, Atlantic salmon (*Salmo salar*) resistance to *C. rogercresseyi* has shown a low to moderate heritability with values between 0.12–0.32 [8,9,10]. Due to the suggested heritability for lice resistance in Atlantic salmon and phenotype variation, a genetic component associated with resistance has been suggested. However, molecular studies that involve the association of functional genomics with Quantitative Trait Loci (QTL) markers have been scarcely conducted [11]. A QTL marker associated with the MHC region is reported for the Atlantic salmon resistance to *L. salmonis* [12]. Meanwhile, QTLs on different chromosomes have been reported for resistance to *C. rogercresseyi*. One of these QTL markers was associated with *TOB1* gene, a transcription factor that regulates T-cell proliferation. A second QTL was associated with *STK17B* gene, related to T-cell apoptosis [13]. In addition, a genotyping study conducted in Atlantic salmon and rainbow trout, a susceptible salmonid species to sea lice infestation, reported that the genetic variation associated with resistance to sea lice infestation is explained by 3% and 2.7%, respectively, in these species [13]. Furthermore, the SPNs on resistant phenotypes were associated with immune response and cell migration [14]. Due to the low heritability of these markers, transcriptional information can improve the selection programs.

Few studies have focused on the transcriptional differences between resistant and susceptible salmon families to sea lice infestation. For instance, Holm et al. (2014) evaluated the transcriptional patterns of Atlantic salmon families categorized as resistant (R) and susceptible (S) to *L. salmonis* infestation. From the RT-qPCR analysis of 34 immune-related genes, differences between families were reported. Therein, the authors highlight the upregulation of Th1- and Th2-related genes in resistant families, while genes such as *MHCII* and *COX2* have been associated with susceptible families [15]. Moreover, Robledo et al. (2018), by RNA-Seq analysis, reported 43 genes differentially modulated in Atlantic salmon families, defined as R and S, after 8 days of *C. rogercresseyi* infestation. Among these genes, the S families showed an up modulation of heme biosynthesis, immune receptors, and muscle contraction-related genes compared with the R families [16]. On the other hand, non-coding RNAs, such as long non-coding RNAs (lncRNAs), with a relevant role in gene expression modulation, have the potential to be used in selection programs. In addition, in mammals, non-coding regions exhibited a high number of SNPs that can be associated with transcript changes [17]. Furthermore, a tissue- and species-specific lncRNAs modulation in salmon species with different susceptibility to *C. rogercresseyi* infestation has been reported by Valenzuela-Muñoz et al. [18].

Recently, we published methods to determine the chromosome gene expression index (CGE), which allows for identifying the transcriptional differences among experimental conditions. Furthermore, this approach considers the chromatin conformation that impacts gene expression and regulation [19]. We hypothesize that the Atlantic salmon genome modulation exhibit differences associated with the resistant and susceptible phenotypes with the potential to be used in the salmon breeding programs to select the phenotype desired. To have a better understanding of Atlantic salmon resistance to sea lice, this study used a whole-genome transcripts profile approach to determine transcriptome differences between Atlantic salmon families denoted as resistant (R) and susceptible (S) to *C. rogercresseyi* infestation and identify new SNPs markers with the potential to be used in salmon aquaculture. 

## 2. Materials and Methods

### 2.1. Experimental Trial

Atlantic salmon post-smolts from 121 families from the genetic program of Hendrix Genetics Aquaculture (year class 2019) were infected with *C. rogercresseyi* in the VESO Chile experimental unit (Colaco, Los Lagos Region, Chile). The Hendrix Atlantic salmon families were obtained according to pedigree information from reproducers from the year class 2025. A total of 2263 fish (18 fish per family) of an average weight of 175.4 ± 25 g were identified using PIT-tags, acclimated in seawater under controller temperature (12.7 ± 1.1 °C), and fed with commercial feed for 15 days. The infestation was performed in two tanks of 4 m^3^ (density of 52 kg/m^3^) for two weeks. Fish were infected with 35 copepodid per fish under the infestation protocol of VESO Chile. After 14 days, Chalimus (Ch II-III) burden was determined by counting all fish. Salmon families were denoted resistant and susceptible to the number of sea lice (Appendix A). For RNA-Seq analysis, skin and head kidney samples were taken from the top 4 families with the lowest (resistant families, R) and highest (susceptible families, S). Samples were fixed in RNA later and stored at −80 °C until total RNA extraction. The study was conducted according to the guidelines of the 3R and approved by the Ethics, Bioethics, and Biosafety Committee of Ethics, Bioethics and Biosafety of the Research and Development Vice-rectory of the University of Concepción, Chile (approval code CEBB1125-2022, April 2022).

### 2.2. High-Throughput Transcriptome Sequencing 

Total RNA was isolated from each experimental fish group using the TRizol Reagent (Ambion^®^, Austin, TX, USA) following the manufacturer’s instructions. The isolated RNA was evaluated by the TapeStation 2200 (Agilent Technologies Inc., Sta. Clara, CA, USA) using the R6K Reagent Kit. Three biological replicates of R and S families were separately sequenced by tissue and sampling point from each experimental fish group. Five individuals were used for the RNA extraction and then pooled for the library preparation for each replicate. Briefly, total RNA was extracted from each individual, five per group, and the RNA pool was prepared using a similar RNA quantity (5 μg.) for each individual. RNAs with RIN > 8.0 were used for double-stranded cDNA libraries construction using the TruSeq RNA Sample Preparation Kit v2 (Illumina^®^, San Diego, CA, USA). Raw sequencing data were deposited on NCBI Sequence Read Archive (SRA) (PRJNA945359). 

### 2.3. RNA-Seq Data Analysis

Raw sequencing reads were assembled to the Atlantic salmon genome (GenBank GCA_905237065.2) using the CLC Genomic Workbench v22 software (QIAGEN, Aarhus, Denmark) for each tissue separately. The assembly was performed with overlap criteria of 70% and a similarity of 0.9 to exclude paralogous sequence variants (Renaut et al., 2010). The settings used were mismatch cost = 2, deletion cost = 3, insert cost = 3, minimum contig length = 200 base pairs, and trimming quality score = 0.05. After assembly, singletons were retained in the dataset as possible representatives of low-expression transcript fragments. Differentially expression analysis was set with a minimum length fraction = 0.6 and a minimum similarity fraction (long reads) = 0.5. The expression value was set as transcripts per million model (TPM). The distance metric was calculated with the Manhattan method, with the mean expression level in 5–6 rounds of k-means clustering subtracted. Finally, Generalized Linear Model (GLM) available in the CLC software was used for statistical analyses and to compare gene expression levels in terms of the log2 fold change (*p* = 0.05; FDR corrected). The metric distance was calculated with the Manhattan method, where the mean expression level in 5–6 rounds of k-means clustering was subtracted.

### 2.4. Whole-Genome Transcript Expression Analysis

Raw data from each experimental group were trimmed and mapped to the Atlantic salmon genome (GenBank GCA_905237065.2) using CLC Genomics Workbench v22 software (QIAGEN, Aarhus, Denmark). Threshold values for mRNAs and lncRNAs were calculated from the coverage analysis using the Graph Threshold Areas tool in CLC Genomics Workbench software. Here, an index denoted as Chromosome Genome Expression (CGE) was formulated to explore the whole-genome transcript expression profiling previously described by our group [19,20]. The CGE index estimation represents the percentage of the mean coverage variation between R and S Atlantic salmon families for the same locus. Briefly, the transcript coverage values for each dataset were calculated using a threshold of 10,000 to 90,000 reads, where a window size of 5 positions was set to calculate and identify chromosome regions differentially transcribed. Finally, the threshold values for each dataset and CGE index were visualized in Circos plots [21]. The contigs sequences obtained from each tissue were blasted to CGE regions to enrich the number of transcripts evaluated by RNA-Seq analysis, as was previously described. In addition, the sequences were extracted from the Atlantic salmon genome near the threshold areas in a window of 10 kb for each transcriptome.

### 2.5. LncRNA Identification and Genome Localization

LncRNAs in R/S Atlantic salmon skin data were identified following the previous pipeline designed by our groups [22]. The identified LncRNAs were mapped against the last version of the Atlantic salmon genome (GenBank GCA_905237065.2). Thus, lncRNAs were mapped using the following parameters: length fraction = 0.8, similarity fraction = 0.8, and mismatch, insertion, and deletion cost of 2, 3, and 3, respectively. The lncRNAs were mapped and annotated in the Atlantic salmon genome. Later, any coding gene flanking up to 10,000 nucleotides from any annotated lncRNA was identified and extracted for further analysis. Functional enrichment analysis of lncRNA-neighbor genes was performed, as was explained before.

### 2.6. Functional Annotation and SNP Identification

Differentially expressed contigs were annotated through BlastX analysis using a custom Atlantic salmon protein database constructed from GenBank and UniProtKB/Swiss-Prot. The cutoff E-value was set at 10^−10^. Transcripts were subjected to functional enrich analysis using g:Profiler [23]. The results were plotted using the cluster profile R package.

SNP identification was performed by mapping the skin tissue reads to the Atlantic salmon genome using the parameters described previously. Then, the SNPs were identified with the variant detection tool available in the CLC Genomics Workbench v22 software (QIAGEN, Aarhus, Denmark). The parameters used were a minimum frequency of 35% and minimum coverage of 10.

## 3. Results

### 3.1. Transcriptomic Profile of Skin and Head Kidney Tissue of Atlantic Salmon R/S Families

From 121 Atlantic salmon families, four Atlantic salmon families were selected, two with the lowest sea lice burden (R) and two with the highest sea lice burden (S) according to the sea lice burden recorder (Appendix A). A PCA analysis of the four families demonstrates a similar differentiation between the R1 and S1 families in the skin and head kidney (Appendix A). Thus, these two families were used for all analyses. The family denoted as R2 was used as a control (reference) for differential expression analysis. The R2 family was denoted as the control group because the PCA analysis showed that this family expression profile is between the R and the S families (Appendix A). 

Whole transcriptomic variation was evaluated in the skin and head kidney tissue. Interestingly, the whole transcriptomic analysis represented in the heatmap of both tissues exhibited differentiation between the R and S families, showing a cluster of transcripts highly expressed R and S (Figure 1A). For instance, cluster 2 of the skin tissue is highly expressed in the R family and associated with genes such as *C-C chemokine receptors*, *coagulation factor V-like*, *collagen alpha*, and *myosin*. While in the S family, the genes upregulated such as *mucin-5B* and *metalloendopeptidase* (Appendix A). In the case of the head kidney, cluster 1, highly expressed in the R family, exhibited genes associated with an immune response, such as *interleukins* and *MHCI*.

Also, variations among the number of transcripts differently modulated in the R and the S families in each tissue were observed (Figure 1B). For instance, the S family presented the highest number of exclusive transcripts differentially expressed (DE) in skin tissue, with 3819 transcripts (Figure 1B). In contrast to head kidney tissue, where the R family exhibited the highest number of exclusive transcripts DE, 7134, compared with the S family, which exhibited 3607 exclusively modulated transcripts and DE in this tissue (Figure 1B). Notably, no shared DEGs were observed between tissues.

Furthermore, from skin-exclusive DEGs, the R family showed a high abundance of genes associated with Molecular Function (MF), such as anion binding, small molecules binding, oxidoreductase activity, protein kinase activity, and cytokine binding, compared with the S family skin. Interestingly, the S family skin tissue presented a high abundance of genes associated with iron ion binding, hydrolase activity, and heme binding compared with the R family (Figure 2). For the head kidney tissue, the GO annotation of DE transcripts allows us to identify more abundance of genes associated with iron binding, heme binding, and cytokine receptor activity in the S family than the R family (Figure 2). In addition, from the KEGG pathway annotation of skin transcripts, the mTOR signaling pathway, MAPK signaling pathway, metabolic pathway, endocytosis, and apoptosis were identified and were more abundant in the R family than in the S family (Appendix A). The differentially expressed genes annotated in both R and S head kidney tissue were associated with the MAPK signaling pathway, metabolic pathway, and cytokine–cytokine receptor interaction (Appendix A). 

### 3.2. Whole-Genome Transcriptome Analysis of R/S Atlantic Salmon Families

The whole-genome transcriptome analysis showed differences between resistant and susceptible families at the chromosome level in both tissues (Figure 3A). For instance, skin tissue data showed the highest number of over-expressed regions in the R family than the S family. Interestingly, eleven chromosomes exhibited differences in chromosome gene expression (CGE) regions up to 60% between R and S families (Figure 3B). Moreover, the highest number of transcripts for the CGE region annotated MF associated with transferase activity, transcription regulator activity, metal ion binding, ion binding, and cation binding (Figure 4).

For the head kidney tissue, eleven chromosomes were observed with a CGE index over 60% from the Atlantic salmon genome (Figure 3B). It draws attention that the transcripts of head kidney tissue from the susceptible family showed the highest expression levels in CGE regions than the resistant family, different from what was observed in skin tissue (Figure 3A). The high number of genes identified in the CGE regions was associated with MF as binding, catalytic activity, ion binding, and metal ion binding. Interestingly, transcripts associated with salmon secretomes, such as protein kinase activity, protein tyrosine kinase activity, and protein serine/threonine kinase activity, were also annotated (Figure 4).

### 3.3. Looking for R/S Transcriptome Differences in CGE Areas of Atlantic Salmon Skin

From the transcripts presented in the CGE area of skin tissue (Figure 5A), it was observed that the S family showed the highest number of transcripts differentially modulated (1080) than the R family (734) (Figure 5B). Furthermore, from the DEGs analysis of CGE genes, an upregulation in the R family related to the S family of genes associated with tissue repairs, such as *myosin* and *collagen alpha*, was observed; it also highlights the up-regulation of immune-related genes such as *immunoglobulin superfamily member*, *TNF receptor*, and *TLR13* (Figure 5C). Interestingly, genes associated with immune response activation, such as *T-cell surface antigen CD2*, *MHC class I*, *B-cell receptor*, and *MMP19*, were down-modulated in the R family compared to the S family.

### 3.4. LncRNAs Identification in R/S Atlantic Salmon Skin

A total of 1830 lncRNAs were identified in the skin tissue of R and S Atlantic salmon families. Of them, 102 lncRNAs were exclusively expressed in the R family and 191 in the S family (Figure 5D). Interestingly, from the evaluation of change expression among lncRNA-neighboring genes presented in both Atlantic salmon families, an upregulation of *mucin-5B-like*, *MCHII antigen alpha chain*, and *myosin-7* has been observed in the R family compared with the S family. While genes, such as *B-cell antigen complex*, *hemoglobin subunit beta*, and *receptor protein–tyrosine kinase*, were downregulated in the R family compared to the S family (Figure 5E).

### 3.5. SNPs Variation in Skin CGE Genes

The highest number of SNPs identified were associated with the resistant family compared with the susceptible, most of them heterozygous (Figure 6A). Interestingly, Chr1 and Chr14 exhibited high SNP frequency in both families. At the same time, the Chr10 showed the lowest frequency (Figure 6B). 

In the resistant family, 7116 SNPs were identified, 3246 with a non-synonymous variation. In the other case, the S family presented 848 SNPs variants in 293 genes, where 428 SNPs are non-synonymous variations. GO enrichment analysis for genes with non-synonymous variation in the R and S families resulted in genes associated with salmon secretome response, such as kinase activity, protein kinase activity, and protein serine/threonine kinase activity, among others (Appendix A). It is noteworthy that the number of exclusive, synonymous SNPs in the R and S families were 460 and 179, respectively (Figure 7A). In the R family, the two most representative MF were associated with ATP-depended activity and protein serine/threonine kinase activity (Figure 7B). Among the MF annotated in the S family genes, the MF phosphotransferase activity and kinase activity are highlighted (Figure 7B). Notably, among genes with SNPs variants in the R family, collagen alpha-1, non-specific serine/threonine protein kinase, tissue factor pathway inhibitor, and MMP19 was observed to be upregulated compared with the S family (Figure 7C). In addition, among genes with an SNP variation in the S family, an upregulation of genes associated with an immune response, such as *interferon-induced protein*, *NF-kappa B inhibitor*, and *MHC-I*, has been observed. In addition, the *mucin 5AC-like* gene associated with mucosal immunity is highlighted (Figure 7C).

## 4. Discussion

The worldwide salmon industry exhibited a considerable challenge due to the prevalence of numerous pathogens spreading during the production cycle. Thus, sustainable strategies have been incorporated to improve production and maintain animal welfare. Using genetic tools to select desired traits such as high growth rate or disease resistance is one of these strategies [24]. However, the selective breeding of salmon families with resistance to sea louse infestation is still challenging due to the low heritability of the genetic markers [7,13]. Thus, it is necessary to improve current selection tools with transcriptional information. Therein, we use genomics tools to identify transcriptomic differences between Atlantic salmon families resistant and susceptible to sea louse infestation to increase the knowledge of molecular processes associated with salmon resistance to sea lice. This knowledge can complement the genetics tools used in salmon breeding programs. 

We use a chromosome gene expression (CGE) index [19] to determine the chromosome regions with high differences between the R and the S families. One of CGE analysis’s advantages is that it allows for determining the transcribed chromosome regions, including non-coding RNAs, and identifying differentially expressed loci [19]. It is relevant considering the number of gene duplications presented in the salmon genome [25]. Notably, the genes located in CGE areas of skin tissue related to Molecular Functions (MF), such as transferase activity, metal ion binding, ion binding, and cation binding, were annotated. In addition, in the head kidney CGE areas, the MF metal ion binding and ion binding were annotated. Notably, the role of nutritional immunity as an Atlantic salmon strategy to respond to sea lice infestation has been previously described [26,27]. Moreover, an over-expression of the *heme oxygenase* gene, associated with iron homeostasis, in Atlantic salmon susceptible families in response to *C. rogercresseyi* has been observed [16]. Upregulation of genes associated with immune response GO terms was reported previously in the study performed in the healthy skin of Atlantic salmon exposed to *C. rogercresseyi* [16]. Our study annotated the immune MF, such as cytokine activity and type I interferon receptor binding. However, the number of transcripts annotated in this MF was less than those associated with the metal/ion binding, which was more abundant in the R family. This suggests a relevant role of nutritional immunity in the Atlantic salmon’s resistance to sea lice infestation.

From a transcriptional study among salmon species with different resistance to sea louse infestation, a relevant role of metalloprotease genes has been observed [28]. In addition, a high presence of protease in Atlantic salmon mucus is described in response to the salmon louse *L. salmonis* infection [29]. In this study, among the CGE areas of the head kidney tissue in the R and S family, a high number of genes with a Molecular Function associated with protein serine/threonine kinase, peptidase, kinase, and metalloprotease activity were annotated, showing the relevance of these molecular functions in the salmon response to the sea lice infestation. 

Due to the relevance of skin tissue as the first immune barrier during sea louse infestation, the study was focused on molecular changes of skin tissue chromosome regions with high differences between R and S families. Notably, immune genes such as *iNOS* and *MHC class I* genes have been reported to be down-modulated in salmon species with high resistance to the salmon louse [28]. Otherwise, comparing the immune response between Atlantic salmon and Coho salmon, these genes have been associated with the Atlantic salmon response to *C. rogercresseyi* infestation [30]. Interestingly, in this study, immune related-genes such as *iNOS*, *MHC class I*, or *MMP19* were down-regulated in the resistant family compared with the susceptible one. Furthermore, similar to [16], the C-X-C chemokine receptor was down-modulated in the R family compared with the susceptible. Besides, the transcriptional analysis describes a high expression of genes associated with muscle contraction, such as *myosin,* in Atlantic salmon skin with susceptibility to sea lice infestation [16]. Furthermore, a QTL candidate has been described for Atlantic salmon with resistance to *C. rogercresseyi* on chromosomes 3 and 21 in the *TOB1* and *STK17B* genes, associated with cell proliferation [13]. In addition, Atlantic salmon with a low resistance to *L. salmonis* has been reported with a thicker epidermis [15]. Moreover, the authors reported an association between the *keratin 8* gene and Atlantic salmon resistance to the salmon louse [15]. In addition, a transcriptomic study in Atlantic salmon fins reported a high abundance of genes associated with tissue repair process during early infestation stages [31]. Here, the Atlantic salmon family with resistance showed an upregulation of the genes related to fish cell proliferation, such as *myosin*, *collagen alpha chain*, and *hemicentin-2*. Thus, it is possible to suggest that Atlantic salmon resistance to *C. rogercresseyi* infestation is associated with the epidermal repair ability of fish as the first barrier than the modulation of immune-related genes.

It has been discussed that in mammals, the main SNPs used for the genome-wide association are located in non-coding regions, which may influence gene transcription [17]. The transcriptomic tools allow us to identify and localize non-coding RNAs in a genome and, in turn, to know their neighboring genes that the lncRNAs can potentially modulate. In this study, we characterized the lncRNA-neighboring genes located in CGE areas. Interestingly, among the neighboring genes up-regulated in the resistant family compared with the susceptible, immune-related genes were identified, such as *interferon-induce protein* and *MHCII*. Notably, the *MHCII* has been associated with salmon species resistant to *L. salmonis* [28]. Moreover, in the comparative study of Atlantic salmon families categorized as low and high resistant to salmon louse infestation, the *MHC class II* expression was associated with low-resistant individuals [15]. In addition, among the lncRNA-neighboring genes, it is also possible to observe genes related to cell proliferation, similar to DEGs previously mentioned in the R family. The upregulation of the *mucin 5B* gene in the resistant compared with the susceptible family is noteworthy. Mucins are proteins associated with the response to injuries and are involved in pathways such as cell proliferation [32,33]. Furthermore, they are an integral part of the mucosal barrier and are essential in mucosal immunity [34]. In gilthead sea bream, its role in the intestine health in response to parasites has been reported [33]. In addition, their putative defensive role has been registered under helminth infestation [35]. This confirms the relevance of triggering tissue repair in the Atlantic salmon skin during the sea lice infestation, suggesting a mucosal immunity mechanism in the Atlantic salmon resistance to the sea louse infestation.

This study also reported a specific SNP variation associated with the R or S family. Notably, from identifying SNP variation in the R and S families, we identify SNP variations in genes associated with tissue repair and secretome response, similar to the DEGs located in chromosome areas with high expression differences between the R and S families and lncRNA-neighboring genes. From an SNP panel for breeding selection, it is desirable to identify non-synonymous SNPs to associate the variation with gene function. For instance, the STK17B gene that was upregulated in Atlantic salmon families with resistance to sea lice also showed a non-synonymous variation in [13]. However, in our study, the specific SNPs observed in the R or the S families were synonymous. Interestingly, this type of SNP, although it does not affect protein translation, has been described to play a role in mRNA splicing, stability, protein structure, and folding [36]. Thus, synonymous SNPs are a marker with a high application for selection programs. Further studies will be conducted to validate the correlation of these SNPs with the R or the S Atlantic salmon families.

## 5. Conclusions

This study reports transcriptional differences at the chromosome level between phenotypes of Atlantic salmon families, evidencing resistance and susceptibility to sea lice infestation. Salmon families exhibited specific chromosome regulation during the infestation. These high presences of genes are associated with immune response and nutritional immunity. Notably, the salmon family with resistance to sea lice infestation responded by activating genes associated with the cell proliferation process. Thus, our results suggest that Atlantic salmon resistance is associated with a high capacity to repair tissue injuries generated by the early infestation stage of sea lice. Further studies will be conducted to validate the tissue-repair-associated SNP variation identified in this study.

## Figures and Tables

**Figure 1 genes-14-00986-f001:**
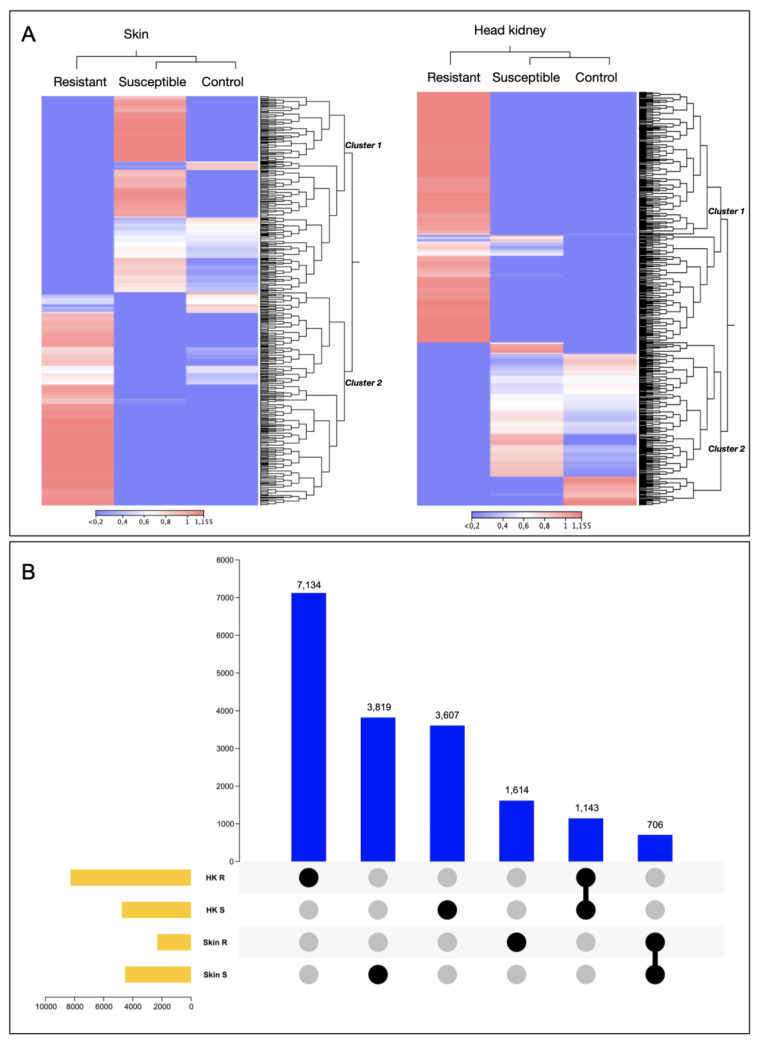
Transcriptomic profile of resistant and susceptible Atlantic salmon families during a *C. rogercresseyi* infestation. (**A**) Heatmap representation of the transcriptomic profile of R/S families in Atlantic salmon skin and head kidney. (**B**) Upset plot of differentially expressed genes of Atlantic salmon R/S families.

**Figure 2 genes-14-00986-f002:**
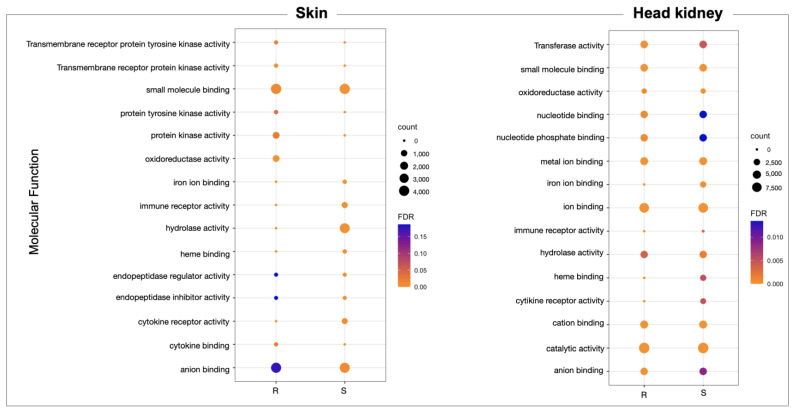
GO enrichment analysis of molecular process differentially expressed in the R and S Atlantic salmon families during a sea lice infestation.

**Figure 3 genes-14-00986-f003:**
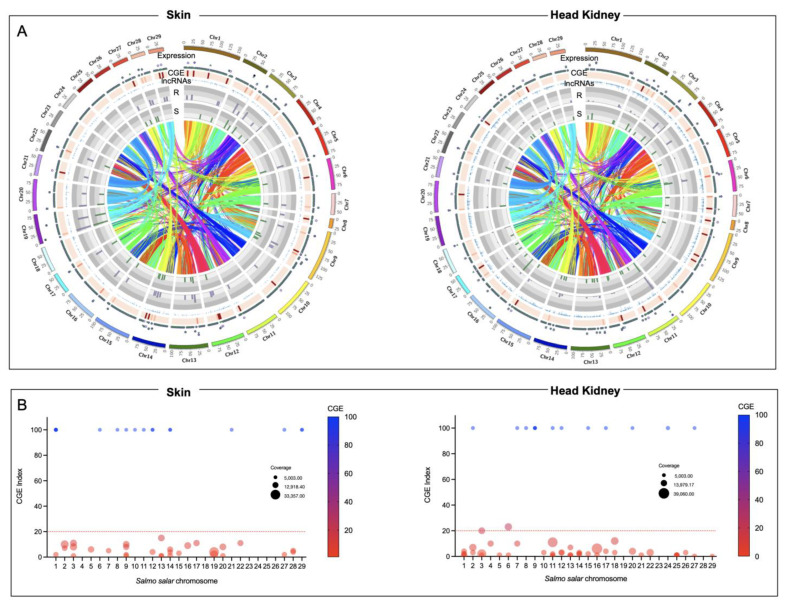
Chromosome gene expression analysis of R/S Atlantic salmon families. (**A**) Circos plot of skin and head kidney tissue. Heatmap in red showed the differences in expression variation between both groups. The scatter plots showed the gene expression levels of the R family in purple and the S family in green. The lncRNA localization is indicated in blue. (**B**) Manhattan plot of CGE index for each Atlantic salmon chromosome.

**Figure 4 genes-14-00986-f004:**
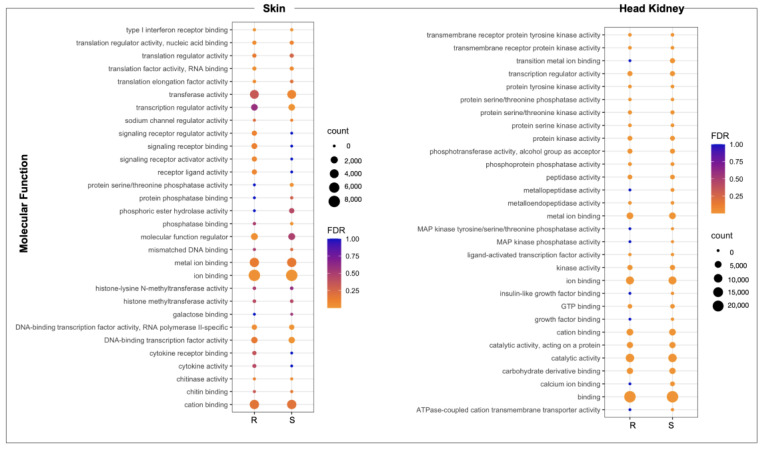
GO enrichment analysis of genes located in CGE areas of skin and head kidney tissue of R/S Atlantic salmon families.

**Figure 5 genes-14-00986-f005:**
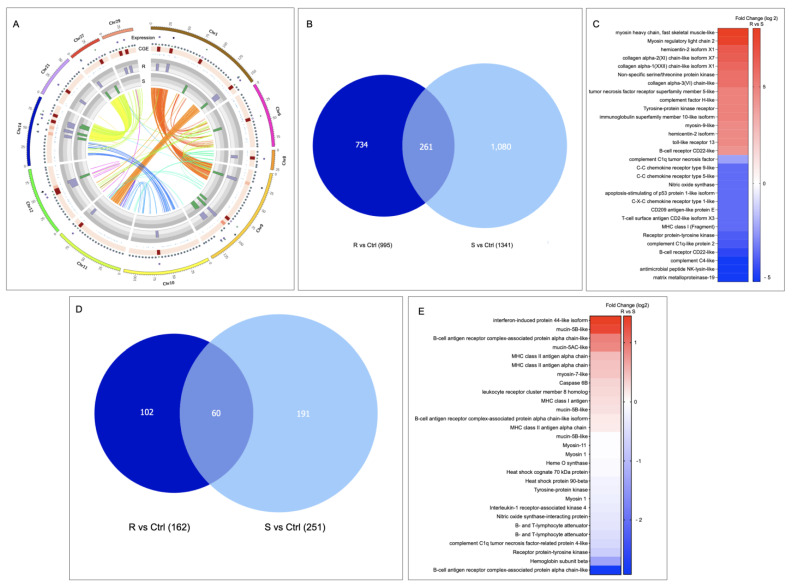
Skin transcriptional differences between R/S Atlantic salmon families. (**A**) Circos plot of chromosomes with CGE index over 60%. (**B**) Veen diagram representation of CGE genes differentially expressed in the R/S families compared with the control family (FC 1.5 *p*-value 0.05). (**C**) Expression analysis of genes differentially modulated in skin tissue. (**D**) Veen diagram representation of lncRNAs differentially modulated in the R/S families compared with the control family (FC 1.5 *p*-value 0.05). (**E**) Expression profile of lncRNA-neighboring genes.

**Figure 6 genes-14-00986-f006:**
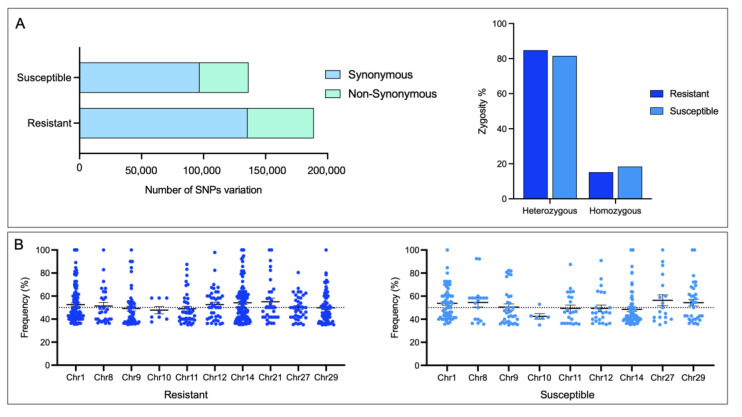
Identification of SNPs in skin CGE genes. (**A**) Number of SNPs and their zygosity in skin of R/S Atlantic salmon families. (**B**) Dot plot of SNPs identified in each Atlantic salmon chromosome.

**Figure 7 genes-14-00986-f007:**
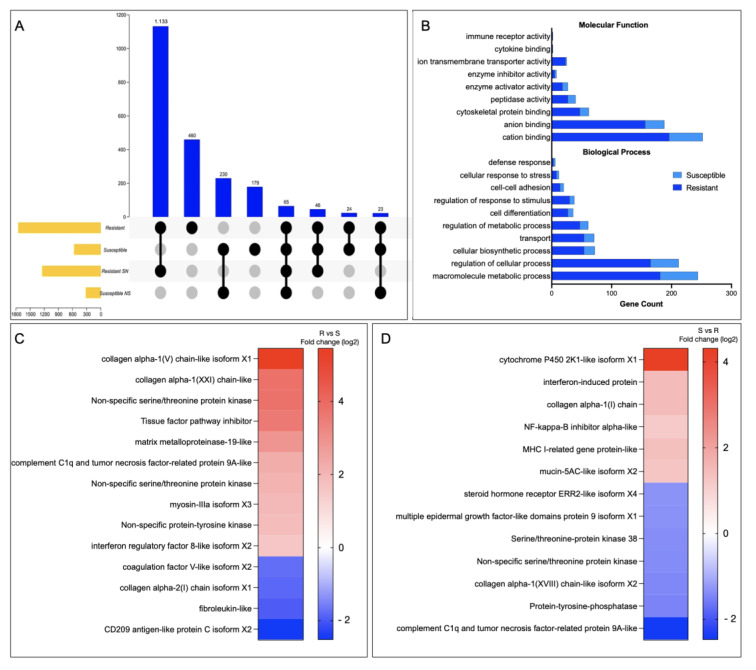
SNPs associate with R/S Atlantic salmon families. (**A**) Upset plot with the SNP number identified in R/S families. (**B**) GO enrichment analysis for SNP variation exclusive in the R and the S Atlantic salmon families. (**C**,**D**) Gene expression profile of genes with SNP variation in the R and the S Atlantic salmon families, respectively.

## Data Availability

NCBI Sequence Read Archive (SRA) (PRJNA945359).

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
