# Peer review of "Transcriptome Signatures of Atlantic Salmon—Resistant Phenotypes against Sea Lice Infestation Are Associated with Tissue Repair"

_genes, 2023, doi:10.3390/genes14050986_

Round 1

Reviewer 1 Report

General comments
- In general, I really appreciate this work with careful design and abundant data. On the other hand, here are some comments to improve the quality of the manuscript. Final remark: this current manuscript needs major review.

Abstract section.
- The present study presents many results that were not described in the abstract. I suggest adding them in that session. We know that the Abstract is limited (in a matter of words), in this case, lines 19 - 25 can be summarized and add more results.
- Conclusion must be made based on the premises described at the final of introduction section. Conclusion strictly related to the results obtained. Depending on the assumptions/suggestions/probability/hypothesis, it can be considered in the discussion. Review the entire manuscript.
- Usually, key words are words that do not contain in the title of the manuscript. Review the entire of manuscript.

Introduction section
- Currently, most of the scientific manuscript are presented as hypotheses to be more attractive and interesting than description of goals. The present manuscript can be presented with hypothesis. We suggest the authors to present this manuscript with more attractive hypothesis and to make the manuscript more interesting.

Material and Methods sections
- Lines 116-118: "The study was conducted according to the guidelines...". Despite presenting this sentence, the present manuscript needs to present a protocol number with this approval or where to access this approval (website?).
- In many of the results presented, the authors mention that there was a difference in the transcriptome between animals/tissues. However, there was no description in M&M of the statistical analyzes applied in the study. Could you describe/provide?

Results section
- Only criticism in the present manuscript. The way of presenting the results has a lot of description of material and methods or justification. In this case, reading becomes very tiring/repetitive. Be objective and describe only the results obtained. Review the manuscript.

Author Response

Dear reviewer 1.

We do appreciate your suggestions to improve our manuscript. Please find below the responses to every question and/or comment that you have made (green highlighting in the manuscripts).

General comments - In general, I really appreciate this work with careful design and abundant data. On the other hand, here are some comments to improve the quality of the manuscript. Final remark: this current manuscript needs major review.

Abstract section.

- The present study presents many results that were not described in the abstract. I suggest adding them in that session. We know that the Abstract is limited (in a matter of words), in this case, lines 19 - 25 can be summarized and add more results.

R. Thanks for your comment. The abstract was improve as your suggest. Lines 19-21; 29-32.

- Conclusion must be made based on the premises described at the final of introduction section. Conclusion strictly related to the results obtained. Depending on the assumptions/suggestions/probability/hypothesis, it can be considered in the discussion. Review the entire manuscript. R. Thanks for your comment. The conclusion was improve as your suggest. Lines 553-561.

- Usually, key words are words that do not contain in the title of the manuscript. Review the entire of manuscript.

R. Thanks. The keywords were revised.

Introduction section

- Currently, most of the scientific manuscript are presented as hypotheses to be more attractive and interesting than description of goals. The present manuscript can be presented with hypothesis. We suggest the authors to present this manuscript with more attractive hypothesis and to make the manuscript more interesting.

R. Thanks for your comment. We include a hypothesis in the introduction. Lines 112-119.

Material and Methods sections

- Lines 116-118: "The study was conducted according to the guidelines...". Despite presenting this sentence, the present manuscript needs to present a protocol number with this approval or where to access this approval (website?).

R. Thanks for the question. We include the information. Line 136-138.

- In many of the results presented, the authors mention that there was a difference in the transcriptome between animals/tissues. However, there was no description in M&M of the statistical analyzes applied in the study. Could you describe/provide?

R. Thanks for the question. In M&M, statistical analysis is mentioned. Please see lines 164-166.

Results section

- Only criticism in the present manuscript. The way of presenting the results has a lot of description of material and methods or justification. In this case, reading becomes very tiring/repetitive. Be objective and describe only the results obtained. Review the manuscript.

R. Thanks for the comment. We improve the results section.

Reviewer 2 Report

Manuscript ID: genes-2283291 

The manuscript “Transcriptome signatures of Atlantic salmon resistant phenotypes against sea lice infestation are associated with tissue repair and secretome genes” aims to explore the whole-transcriptome changes in Atlantic salmon families with contrasting resistance phenotypes against lice (Caligus rogercresseyi) infestation, using genomics tools to identify transcriptomic differences between Atlantic salmon families resistant and susceptible to increase the knowledge of molecular processes associated with salmon resistance to sea lice.

I truly believe the study is very relevant for animal immunology and aquaculture, however some effort has to be done in order to improve the quality of the manuscript and proceed to publication.

Overall I think one of the most important changes to be done are related to better clarify and describe the experimental design/experimental groups and due comparisons. How were the samples pooled? How were they selected? Explain salmon family concept, among other details.

Also, further describe the analytic method based on "chromosome gene expression" on certain chromosome regions where the differentially expressed genes show more differences between groups. It is a different approach than conventional RNAseq studies that focuses on DEGs on the whole genome?

The results are confusing. Please, improve the way the data are presented; always following a certain order, or between comparison or between tissues, and keep that pattern along the text. Sometimes the text is a bit repetitive and it is not the easiest to read and follow.

How was the rationale behind choosing 14 days of infestation? Is it related to the life cycle of the parasite?

The scientific name of the salmon was not mentioned once.

Other noticed details include:

Introduction

Gene abbreviation in italic (e.g., TOB1, STK17B, MHCII, and COX2, for example).

Materials and Methods

How was the density of the fish in the tanks? Two tanks of 4 m3 is enough for 2,263 fish? 

“After 11 days, Chalimus (Ch II-III) burden was determined by counting all fish. Finally, skin and head kidney tissue samples of the top 4 families with the lowest (resistant families, R) and highest (susceptible families, S) sea lice burden were taken.” In the abstract authors mentioned 14 days of infestation, it should be mentioned clearly in this topic of the M&M, please, since it is not clear up to what day the infestation was maintained. 

What is the size of the “Atlantic salmon families”? Or how are the animals organised within these families?

“Three biological replicates of R and S families were separately sequenced by tissue and sampling point from each experimental fish group. Five individuals were used for the RNA extraction and then pooled for the library preparation for each replicate.” I understand from this part that there are three replicates with a pool of five fish each. What I do not get is how does it relate with the top two lowest (R) and highest (S) infested families? Are they separate groups? The experimental groups should be clarified.

Please confirm in the “RNA-seq data analysis” topic if the p = 0.005 ou would that be 0.05?

LncRNA and functional annotation and SNP identification were performed just in the skin tissues? (described in the end of material and methods)

Results:

Line 207: Figure 1A caption does not mention head kidney, please include.

Line 194: “The family denoted as R2 was used as a control (reference) for differential expression analysis.” I fail to understand how the control group was set?! Does it also participate in the infection experiment? This is crucial to understand the comparisons.

Line 196: “Interestingly, the heatmap representation of both tissues exhibited a clustering between control and susceptible groups”. Personally I disagree. I do not see a full correlation/clustering of most of the DEGs in those groups, just part (maybe ⅓). What do authors mean when referring to clustering between control and susceptible? Besides, still I believe it may not be clear to the reader how the control group was treated.

Line 228: In the figure 2 caption please check if the term would be “molecular process” or “molecular function”, which is a class of GO. Why did the authors not present comparative enriched terms for Biological Process? It would be convenient to increase the size of the GO terms in the figure 2. 

Line 232: “Transcriptome modulation of R/S families was evaluated at the genome level, to determine the differences between each family at the chromosome level”. What do authors infer when saying “at the chromosome level”? For me the transcriptome analysis will capture molecular changes at the gene expression level; I do not see how that correlates to specifically chromosome (Repeated in line 235). Is it an analysis of the gene expression from certain chromosomes or chromosome regions? Please, clarify.

Figures 5 and 7 should be amplified.

Discussion

How could the present research provide insights for Genomic Selection (GS) programs aiming to work on creating better lineages that would be more resistant to infections?

Line 345: “In this study, we performed a chromosome-scale gene expression analysis to determine the genomic regions exclusively modulated in R or S families for biomarkers prospection.” I guess this explanation, maybe on methods, about the type of analysis may be crucial for the better understanding of the results. So, briefly introduce this “chromosome-scale gene expression analysis” before (I’ve done a similar comment above).

I guess the readers would further benefit from an analysis of the pathways more impacted by the differences in infection rather than particular genes that have been affected. Of course, after a broader analysis some genes can be picked and described, but I think it is more constructive to explore pathways and processes than pointing genes that are up or down-regulated (e.g., lines 372-380). In this sense, I miss some elaborate discussion on the enriched GO terms presented in the results (for example here: lines 239-240); their biological relevance and interaction should be explored.

Comparing conclusion and title I was wondering if it isn't a bit forcing the idea of tissue repair and secretome genes, since one can see many other processes have diverged between S and R “families”. If the authors decide to maintain the main conclusion/findings as the title I guess the results and discussion should be more directed and further elaborated on those processes, more than cherry picking DE genes that contribute for those processes. Since the functions pointed in the title now, although very relevant, are not as slashing as they look, maybe a more generalistic title would fit better.

Unfortunately, I had not checked the supplementary material mentioned in the text to conclude this analysis. Eventually, for a second round of review I would like very much to check that material.

Author Response

Dear reviewer 2.

We do appreciate your suggestions to improve our manuscript. Please find in the attached file the responses to every question and/or comment that you have made. (light blue highlighting in the manuscript)

Reviewer 3 Report

It is well written manuscript. I have several minor comments below.

Introduction:

Line 59: Authors should write the abbreviation of Lepeophtheirus salmonis in parenthesis.

Line 66: Authors should write the full name of MHC first, then it can be used as abbreviation.

Line 87: Authors should delete “with” and write “to”.

Results

Line 210: Authors should add “head kidney” after skin.

Figure 2 and 4: It is not easy to read the labels on the figure 2. Authors should make it bigger.

Figure 3B: There are two figures, but it is not shown R or S groups on the figures.

Figure 5: It is not easy to read the labels on the figure 2. Authors should make it bigger on Figure 5B-C-E.

Figure 7: it should be bigger. It is not easy to read and follow.

Author Response

Dear reviewer 3.

We do appreciate your suggestions to improve our manuscript. Please find in the attached file the responses to every question and/or comment that you have made. 
